# Effects of Acute Stress on Psychophysiology in Armed Tactical Occupations: A Narrative Review

**DOI:** 10.3390/ijerph19031802

**Published:** 2022-02-05

**Authors:** Mark D. Stephenson, Ben Schram, Elisa F. D. Canetti, Robin Orr

**Affiliations:** 1Faculty Health Science and Medicine, Bond University, Robina 4226, Australia; bschram@bond.edu.au (B.S.); ecanetti@bond.edu.au (E.F.D.C.); rorr@bond.edu.au (R.O.); 2Tactical Research Unit, Bond University, Robina 4226, Australia

**Keywords:** arousal, performance, pressure, stress response

## Abstract

The ability to perform under extreme pressure is one of the most sought-after qualities in both sports and tactical (military, law enforcement, fire, and rescue, etc.) occupations. While tactical performance relies on both physical and mental capabilities to achieve a desired outcome, it is often hampered by the stressful environments in which these personnel work. The acute stress experienced by tactical personnel can interfere with occupational performance, impacting both physical execution of tasks and decision-making. This narrative review discusses the implications of acute stress on the psychophysiology and physical performance of personnel serving in armed tactical occupations.

## 1. Introduction

One of the most sought-after qualities in armed tactical occupations (military and law enforcement) is the ability to perform in stressful situations and under stressful conditions [1,2]. Armed occupations in the tactical field often require individuals to apply technical and tactical skills to difficult tasks where the outcomes could have extreme consequences [2,3]. The high pressures experienced by personnel serving in tactical occupations requires the ability to perform tasks in the presence of high physiological arousal and psychological stress [2,4,5]. Both military and law enforcement personnel often face an occupational environment that is influenced by sociopolitical opinions, media, public perception, and the threat of daily violent encounters, all of which accumulate into occupational stress [3,6,7].

Tactical occupations, in general, are charged with public safety and have the responsibility to protect citizens and their property [8,9,10,11,12]. For those in armed tactical occupations, meeting this responsibility sees personnel at every level of military and law enforcement exposed to the chronic stressors associated with physical harm, social unrest, shiftwork, and in many cases, lack of organizational resources [13,14]. In law enforcement, these stressors are concomitant to those faced in general duty policing activities such as effecting an arrest, crowd control, and attending domestic disturbances [10]. Military special operation forces (SOF) or law enforcement special response teams (SRT) or special weapons and tactics (SWAT) teams are further exposed to higher-risk situations, often involving life and death decisions [15]. These stressors can be further compounded by the media’s messaging of specific incidents, influencing the public’s view of security forces or law enforcement and their actions [6]. In law enforcement, personnel are professionally trained in their response to deadly force, most notably when it is perceived to be life threatening to the public or themselves, and any misperceptions can have catastrophic effects [16]. However, this constant threat of violence, lack of public support, and exposure to tragic events influences the state of vigilance that a police officer is constantly under [13]. 

Like law enforcement, military security or combatant operations involve a high level of uncertainty and require immediate and appropriate action in an acute crisis situation [17,18]. The enduring deployments and unpredictability of direct action creates a high stress environment, significantly increasing the demand of cognitive resources from soldiers, placing them at a greater risk of errors and mishaps in the field [18,19]. This increase in cognitive demand can influence combat performance, resulting in poorer strategic decision-making and decreases in shooting accuracy [20]. Military personnel must be able to perform multiple tasks simultaneously or in concert with other soldiers (e.g., a combat medic performing first aid while infantry soldiers are engaged in combat) while under high levels of physical and mental strain, which influences decision-making as well as fine motor function [21].

The impact of pressure (time, urgency, or outcome) on skill-based tasks can be both beneficial and detrimental to performance. Such impact is dependent on the circumstances related to the situation (e.g., social environment) as well as the performer’s association with the task being executed [22]. Acute stress that is brought upon by a sense of urgency or a consequential outcome greatly influences performance [23]. As such, it can result from the desire of an individual to perform at their best, especially where high level performance is deemed critical to desired outcomes (e.g., life-threatening emergencies) [24]. These stress-induced anxieties can result in shifting attention to worries of outcome and thereby impede performance [3,23]. 

Critical situations can impact the individual’s perception of external factors (threat, social unrest, time, etc.), which elicits an emotional condition deemed unpleasant or “toxic” [25,26] and is only manageable when the resources are available for modulation, either through downregulation or upregulation of the response [26]. The use of breath work to assist in this modulation has been very popular as one of those resources available due to its ease of implementation and skill [27,28,29]. Breathing techniques such as diaphragmatic breathing, tactical breathing, or box breathing are more popular with tactical personnel [29].

Performance in high pressured situations is relevant in tactical occupations and can have great social, political, and moral implications based on their outcome [22]. Understanding the relationship between acute stress and performance, as well as the mechanisms behind the stress response, is critical to employing effective interventions and strategies to help regulate the effects. Thus, the purpose of this narrative review was to (1) review the underlying concepts of stress and the stress response and (2) review the concepts of performing under pressure from a skill task perspective. 

## 2. Defining Stress

Armed professionals are exposed to a myriad of stressors while performing their normal duties. Of particular concern are those encounters that require use of force or life and death decisions in which immediate, unplanned action is required. These life-threatening encounters require an immediate and appropriate action, avoiding unintended consequences (e.g., loss of life) [8,9,10,24]. Stress among armed tactical professionals can be in many forms, such as accumulative stress, public perception, social, environmental, and threat [3,6,7,13,14]. 

Hans Selye originally used the term “stress” in the 1930s to describe circumstances where an organism does not respond appropriately to a physical challenge [30]. Selye goes on to describe this adaptation process, the “General Adaptation Syndrome”, as occurring when an organism adapts to stressors, both physically and psychologically [31]. A more modern interpretation of stress is all-encompassing of life’s stressful events, chronic and acute stress experiences, occupational stress, and the psychological and physiological stress reactions [32]. Stress implies that there is a disruption to the otherwise normal environment or an imbalance [32,33,34,35], resulting in an emotional response leading to physiological and psychological hardship [36]. Stress, being a non-specific response to a stressor [31], is also relative to the individual’s perception of what is deemed a “stressor” [32] and is therefore also a non-specific threat or danger [31]. Stressful experiences can be classified as “good”, “tolerable”, or “toxic”. This classification is relative to the individual’s ability to control or modulate the stress response and is dependent on the availability of resources to manage the experience [26]. Selye believed that stress is necessary and is always present, be it at varying levels, and that to be free of stress is the equivalent of being “dead” [32,37]. 

## 3. How the Stress Response Works

When encountering an immediate threat or crisis, these armed tactical professionals experience acute stress resulting in a psychophysiological response [5,8,10,20]. The ability to regulate these responses allows for the armed professional to respond to the threat more appropriately, taking appropriate action [1,2,3,4,5]. The perception of threat or urgency initiates a response with cascading effects. In other words, the brain’s perception is the body’s reality. 

The stress response involves a cascade of adaptive neurophysiological responses initiated by the brain and periphery, through the autonomic nervous system (ANS) and hypothalamus–pituitary–adrenal (HPA) axis [32,34,38]. The two main components to the stress response are the sympathetic–adrenal–medullary (SAM) and the HPA axis, with the SAM primarily involved in the acute response and the HPA axis responsible for long-term defence [34,39]. 

### 3.1. Autonomic Nervous System

The ANS is part of the efferent division of the peripheral nervous system responsible for regulating both smooth and cardiac muscle contraction as well as glandular and organ functions, in order to maintain homeostasis in the face of fluctuating internal and external stimuli [18,40,41]. Functioning continuously with automaticity [40], the ANS is responsible for controlling the stress response by modulating its two branches: the sympathetic and parasympathetic branches [36,42,43,44,45]. The sympathetic branch is often linked to the utilization of energy, whereas the parasympathetic branch is often linked to the “vegetative” functions or rest and recovery [40,44]. During stress, such as a perception of threat, the sympathetic branch of the ANS is upregulated, leading to an increase in heart rate, blood pressure, and respiration [20,34,36,43]. When the perceived threat is no longer there, the parasympathetic branch increases in activation and regulates the physiological responses back to pre-threat levels [36].

Both sympathetic and parasympathetic branches are continuously active, providing input at all times by either increasing or decreasing in activation [41]. This dynamic relationship between the sympathetic and parasympathetic, often referred to as ANS balance, has been linked with good cognitive function [42]. The dynamic balance between these two systems has an effect on the consistency of time between heart contractions or inter-beat-intervals (IBI), which is calculated using heart rate variability (HRV) [46]. Disruption of this balance over the long-term can lead to negative health consequences such as hypertension, cardiac disease, respiratory illness, etc. [32,47]. 

### 3.2. Neuroendocrine Response

The SAM system is responsible for the initial shock response, activating the sympathetic ANS, producing the “fight or flight” response [34]. Once initiated, The HPA axis is responsible for the release of three specific hormones: corticotrophin-releasing factor (CRF), adrenocorticotropic hormone (ATCH), and cortisol [39,45,48]. There are numerous factors that can result in the initiation of the HPA axis (e.g., cold, shock, stress, etc.) [48], including the perception of threat, which is routinely experienced by tactical personnel. Upon perception of threat, the HPA axis initiates the release of a cascade of hormones throughout the body. The hypothalamus releases CRF, which stimulates the release of ATCH from the pituitary gland, stimulating the adrenals to release cortisol into the blood [39,45]. 

### 3.3. Chronic vs. Acute Stress

Stress can be further broken down into “chronic” or “acute” [33]. Chronic stress refers to long-term disruption of homeostasis, resulting in behavioural and health issues [49], whereas acute stress is short-term and does not affect health [32,33,34,35]. Acute stress results from pressure and anxiety induced by external stressors such as time, urgency, and outcome and can affect performance both physically and mentally [2,3]. 

The terms ‘stress” and “pressure” are often used interchangeably throughout the literature when discussing performance. It is worth noting that some researchers attempted to distinguish between the two, describing pressure as having more of an associated effect on performance rather than well-being, whereas stress has an associated affect with well-being rather than performance [50]. They go on to explain that pressure can be thought of as occurring when the outcome results in a consequence such as winning or losing or living or dying or where monetary incentives are impacted [22,50]. The focus on outcome leads to an increase in anxiety and perceived stress, hampering performance [23]. A schematic diagram of the various components impacted by the psychophysiological effects of acute stress is displayed in Figure 1. Unfortunately, there is no clear distinction when alluding to implications on performance in sport and skill [50], making both terms acceptable. For the purpose of this review, maintaining consistency with the existing literature, stress and pressure will be used interchangeably. 

## 4. Acute Stress and Performance

Performance-based tasks are affected by bouts of acute stress, especially those that may result in negative consequences due to less than desirable performance, most notably when these tasks are time-sensitive and involve social constraints resulting in pressure [22]. The pressures experienced during performance often involve an element, or a combination of several elements, that increase the perceived importance of achieving a positive outcome [51]. These types of pressure are often experienced during tactical performance tasks. Occupations involving high-threat situations, which place personnel in harm’s way with uncertain consequences (e.g., police officers), tend to elicit the highest stress response, triggering a cascade of physiological processes [3,39]. For example, general policing duties may involve repeated tasks under various conditions while experiencing psychological stress, such as perceived threat [3,49]. When a situation is perceived to be a high threat (e.g., an officer confronting an uncooperative assailant), a physiological stress response is activated [39], thereby creating conditions of acute stress. Thus, for a police officer, acute stress could result from situations where split-second decision-making is needed and where outcomes can have second- and third-order consequences (e.g., injury, escape, use of lethal force, etc.) [4,10,50]. This acute stress can affect both physical and mental performance. The ability to make appropriate and accurate decisions under these acute stressful conditions, particularly during a critical incident, is essential if an officer is to achieve the desirable outcome [25]. 

Unlike sporting environments, high-risk tactical occupations face the potential of fatal outcomes in the event of failure [2]. Maintaining control of one’s physiology during an extreme pressure event is critical, and the ability to modulate stress and regulate emotions is imperative when it comes to executing technical skills and appropriate decision-making [1,2,50]. Police performance often requires officers to have the ability to respond appropriately to critical incidents, have situational awareness, use verbal and non-verbal communication effectively, make decisions (shoot/no shoot), and exhibit self- and situational- control [3]. These tactical and technical procedures are affected by cognitive anxiety and influenced by the levels of physiological arousal, where when both cognitive anxiety and physiological arousal are high, performance often suffers [52]. Cognitive anxiety in high-pressured environments often leads to attention being drawn outwardly, away from the given task, causing a deterioration in performance [10,23]. Those who are unable to maintain control and modulate both physiological and psychological states influenced by high-pressure situations often perform below their appropriate skill level [50], an outcome commonly termed “choking” [23,24].

## 5. Impacts of Stress

It is well-established that the ability to perform work is not just due to physiological factors such as cardiovascular function but primarily from central nervous system arousal [5]. Stress is an intricate response causing changes in behaviour and variations in both allostasis and homeostasis, affecting the physical and psychological status [36,43,53]. The effects of stress can negatively impact both the cognitive processes of decision-making (leading to more cognitive errors, increased risk taking, poor situational judgement, and possibly an increase in stereotyping and bias) and motor performance [49,54]. 

### 5.1. Stress and Cognition

Acute stress impacts the prefrontal cortex (PFC), influencing core executive function processes (considered higher order cognitive functions) [55]. These consist of working memory, cognitive flexibility, and inhibitory control, with the latter being most impacted by psychosocial stress [39,55]. Cognitive overload and/or fatigue leads to a deterioration in tactical performance [10,56], and it is often the result of two key stressors impacting decision-making, these being time-sensitivity and information overload [54]. For police officers, these stressors can complicate one of the most crucial decisions an officer may have to make, a decision with the potential for the most catastrophic consequences, being the decision to shoot or not to shoot [57]. 

Decision-making is a cognitive function that is influenced by psychological stress [54,55]. As the level of psychological stress increases, the quality of decisions being made degrades [49,54]. Although there are several decision-making models represented in the scientific literature, the underlying premise that is common to all is that decision-making is a sequential process involving the collection of information, estimations of the outcome, deliberation of the information, and selecting the appropriate decision [54,58]. As a higher-level cognitive process, decision-making involves psychophysiological bidirectional feedback between the central autonomic network and the ANS [59]. For example, the act of shooting a firearm involves a cascade of cognitive processes, such as target acquisition, threat or no threat determination, assessment of collateral risk (knowing what is beyond the target), and discharging the firearm [25]. Finally, adding to this occupational stress are the psychosocial and physiological stressors experienced by officers that can vary depending on the geographic location or sociopolitical environment at the time [6,7]. Thus, due to the impact of acute stress on police officers, their decision-making ability is greatly hampered [57], thereby placing the officers’, suspect’s, and citizens’ lives at risk. 

### 5.2. Stress and Motor Performance

Law enforcement duties often require high levels of motor skills to enact police-related tasks such as high-speed driving, physical restraint, and weapons (lethal and less-lethal) manipulation. When performed under acutely stressful situations, these can be susceptible to errors resulting in long-term physical and mental implications [45]. It is well-established in the literature which investigates the effects of pressure and anxiety on performance, specifically skill-based tasks, that performance declines in high pressure situations [22]. Left unmanaged, acute stress can have undesirable effects on task performance such as task execution [60]. Those who are unable to use effective coping skills during high pressure events are not likely to modulate the psychophysiological states necessary to avoid underperforming [50]. 

## 6. Stress and Tactical Performance

Problem solving, reasoning, and planning are critical components to tactical performance and can be greatly impacted by acute stress [39]. Tactical occupations that involve force-on-force encounters or the lethal use of force decisions (e.g., military, police) involve high-level motor performance skills, ranging from physical restraint to weapons manipulation [45]. As such, for tactical personnel, psychophysiological performance can be greatly impacted by their highly stressful work conditions [20,45,61]. The use of deadly force decisions is influenced by both psychological and physical stress, leading to shooting performance (i.e., accuracy, false positives, and faster reaction times) decrements [45] that can lead to negative consequences, such as loss of life [1,3,10]. The literature also substantiates that increases in anxiety also inhibit performance (i.e., shooting) by increasing avoidance behaviours (e.g., blinking, initially turning away) and making it more difficult to address threats appropriately and engage them accurately [10,20]. 

### Stress Modulation

The ability to self-regulate under stressful conditions is vital to those in armed tactical occupations, such as the military or law enforcement, whose reliability depends on performance and sound judgement when executing their duties [50]. Emotions have been shown to play a pivotal role in task performance, affecting cognitive functions and motor performance via pleasant (happiness or desire) and unpleasant (anger or fear) experiences [62,63]. Emotions or high levels of arousal that go unregulated can lead to performance decrements in fine motor control (e.g., trigger control, sighting, and target acquisition) due to higher levels of muscle tension [64], thus placing the officer or soldier and the public in danger. As such, the ability to self-regulate in stressful situations can be critical to tactical task performance and outcomes. 

## 7. Summary

The volume of research supports that performance declines under stressful conditions. The ability to perform occupational tasks under highly stressful and uncertain conditions is critical to success in tactical occupations, such as military and law enforcement. Acute and chronic stress can have a negative impact on tactical performance, decreasing accuracy in shooting, increasing false positives in decision-making, decreasing inhibition (shoot/no shoot), and increasing reaction-time. The decision to use deadly force in an acute critical situation is impacted by psychophysiological stress and can lead to tactical performance decrements and even unintended loss of life.

Occupational duties that expose personnel to scenarios of escalating threat often lead to an increase in cognitive load, resulting in an increase in cognitive errors and a degradation in decision-making quality. To avoid underperforming or negative consequences, the modulation of psychophysiology is critical. Active coping strategies, such as breath work, can help mitigate the detrimental effects and, in some instances, improve performance under pressure. 

## Figures and Tables

**Figure 1 ijerph-19-01802-f001:**
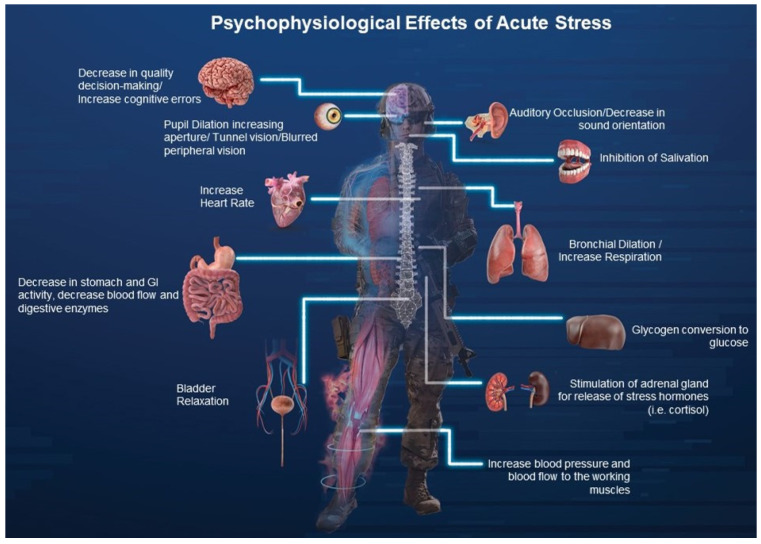
Example layout of the relations between various organs throughout the body and the stress response.

## Data Availability

Not applicable.

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
