# Peer review of "Effects of Acute Stress on Psychophysiology in Armed Tactical Occupations: A Narrative Review"

_ijerph, 2022, doi:10.3390/ijerph19031802_

Round 1
Reviewer 1 Report
The review explored how stress affects tactical occupations. The argument is that the presence of acute stress affects the performance, decision making and effectiveness of the armed personnel. This review has the potential of helping us understand the challenges faced by the tactical occupations and how best we can design interventions to mitigate the identified challenges.
- There is a need to operationalize the tactical occupations since this bit forms the basis of their review. There was some elaboration of this issue but I think the authors can further augment their assertions more concretely. For instance, the number of armed personnel present to secure a country in contrast to the citizen population, their job description, and the likely daily on-the-job stressors they face. The passion of such work includes securing the state and distress for example leaving home and not being sure that one will return or watch people die and they have no way of saving them just like in an accident scenario.
- Task and skill-based performance that is routine could be explained further and how these are actual sources of stress.
- Have a brief introduction and then the stressors included in the introduction can be included in the definition of stress and stress response segments correspondingly. Here we can understand the real stressors associated with the tactical occupations and how these can impair performance in the real-life setting.
- Could you explain more about visual reality environments and debriefing as the other stress inoculation techniques?
- Include more examples of the tactical occupations apart from the police force. How about other armed personnel like border guards, and/ or law enforcement officers?
- Crosscheck the reference list to ensure consistency especially in capitalization, journal article numbers, and volume numbers among others.
Reviewer 2 Report
This review needs rewriting because of 3 reasons:
1) the authors of review write about few professions in which stress is the part of the job in the Introduction, but they really talk about one kind of profession in the next parts of manuscript. I propose to write about one profession and mechanisms which can affect the behaviour of person who is working in this profession. There are no possibility to reveal all mechanisms deeply when you are talking about different professions because then you need to talk about different aspects of the profession.
2) In Introduction the authors write about 3 purposes of the manuscript. I propose to refuse purpose no. 3 or to widen the manuscript writing about different kind of stress regulating mechanisms.
3) the main problem for me is that I do not understand what mission has this review, for what purpose the authors write it, what new they will say. I wish that the authors will think about that, will revise the manuscript seeking to deepen the content of manuscript and make it more one direction.
Reviewer 3 Report
The narrative review manuscript was well written, clear and easy to understand, with suitable reference citations. I have several comments and suggestions for improving the current article.
- Are there any review strategies applied for this article? Please add a separate section if available.
- Please provide the most updated definition of "stress" in the section "2. Defining Stress" with corresponding reference.
- In the section "7. Summary", please remove the citations. The references in this section are not neccesary.
- The authors need to illustrate a FIGURE showing effects of acute stress on osychophysiology in tactical occupations.
Round 2
Reviewer 2 Report
Thank you for corrections which you have made. Now the manuscript seems in better quality.